# Roles of OX40 and OX40 Ligand in Mycosis Fungoides and Sézary Syndrome

**DOI:** 10.3390/ijms222212576

**Published:** 2021-11-22

**Authors:** Yuki Kawana, Hiraku Suga, Hiroaki Kamijo, Tomomitsu Miyagaki, Makoto Sugaya, Shinichi Sato

**Affiliations:** 1Department of Dermatology, Graduate School of Medicine, the University of Tokyo, 7-3-1, Hongo, Bunkyo-ku, Tokyo 113-8655, Japan; yuki_to_bau@yahoo.co.jp (Y.K.); theblankbluesky@gmail.com (H.K.); satos-der@h.u-tokyo.ac.jp (S.S.); 2Department of Dermatology, St. Marianna University School of Medicine, 2-16-1, Sugao, Miyamae-ku, Kanagawa 216-8511, Japan; asahikari1979@gmail.com; 3Department of Dermatology, International University of Health and Welfare, 4-3, Kouzunomori, Chiba 286-8686, Japan; sugayamder@gmail.com

**Keywords:** mycosis fungoides, Sézary syndrome, cutaneous T-cell lymphoma, OX40, TNF receptor superfamily

## Abstract

Mycosis fungoides (MF) and Sézary syndrome (SS), the most common types of cutaneous T-cell lymphoma (CTCL), are characterized by proliferation of mature CD4+ T-helper cells. Patients with advanced-stage MF and SS have poor prognosis, with 5-year survival rates of 52%. Although a variety of systemic therapies are currently available, there are no curative options for such patients except for stem cell transplantation, and thus the treatment of advanced MF and SS still remains challenging. Therefore, elucidation of the pathophysiology of MF/SS and development of medical treatments are desired. In this study, we focused on a molecule called OX40. We examined OX40 and OX40L expression and function using clinical samples of MF and SS and CTCL cell lines. OX40 and OX40L were co-expressed on tumor cells of MF and SS. OX40 and OX40L expression was increased and correlated with disease severity markers in MF/SS patients. Anti-OX40 antibody and anti-OX40L antibody suppressed the proliferation of CTCL cell lines both in vitro and in vivo. These results suggest that OX40–OX40L interactions could contribute to the proliferation of MF/SS tumor cells and that the disruption of OX40–OX40L interactions could become a new therapeutic strategy for the treatment of MF/SS.

## 1. Introduction

Mycosis fungoides (MF) and Sézary syndrome (SS), the most common types of cutaneous T-cell lymphoma (CTCL) [1], are characterized by proliferation of mature CD4+ T-helper cells [2]. MF typically presents in the form of skin patches and/or plaques, which can progress to skin tumors, with subsequent involvement of lymph nodes, peripheral blood, and visceral organs. In some MF cases, skin lesions become confluent and finally develop into erythroderma without blood involvement. SS is defined by the triad of generalized erythroderma, lymphadenopathy, and circulating atypical T cells [1]. Patients with advanced stage MF/SS have poor prognosis, and currently, there is no curative treatment for these patients [3].

Treatment is performed according to the stage, and the main treatments for early MF are topical steroids and ultraviolet light therapy. Erythema and plaques resistant to topical steroids and light therapy require oral medications. Recently, bexarotene has been used worldwide as the first option for both early and advanced MF/SS. In addition, anti-cancer drugs such as histone deacetylase inhibitors, mogamulizumab, gemcitabine, and multidrug chemotherapy are other options for advanced MF/SS. Hematopoietic stem cell transplantation is the only treatment that can be expected to have long-term remission for advanced MF, but about half of the cases die within one year due to recurrence, graft-versus-host disease, or infection [4]. Currently, none of the above existing treatments have clear evidence to drastically improve the prognosis. A considerable number of patients die within a few years, after the diagnosis is confirmed. Therefore, there is an urgent need to elucidate the pathophysiology of MF/SS and to develop new therapeutic agents that target mechanisms different from existing treatments.

It is already known that cell-to-cell interactions between tumor cells and interactions between tumor cells and the tumor microenvironment by autocrine or paracrine signaling contribute to survival and growth in MF/SS. Various cytokines, such as IL-13, IL-15, and IL-32 [4,5,6], and surface proteins, including CD47, CD40, and CD28, provide the direct molecular bridge between tumor cells and adjacent cells, resulting in tumor progression [7,8,9,10]. Here, OX40, also known as CD134, is a member of the TNF receptor superfamily and is a type II transmembrane protein expressed on activated T cells, natural killer cells (NK cells), and regulatory T cells (Tregs). OX40 was first reported as a surface protein of activated CD4-positive T cells in 1987 [11]. Later, it was reported that OX40 was also expressed on T cells locally invading tumors of malignant melanoma and head and neck cancer in 1997 [12,13]. In addition, the anti-OX40 agonistic antibody inhibited the induction of IL-10-producing Tregs and maintained the proliferation and function of effector T cells using peripheral blood in healthy subjects [14]. A study using a mouse model showed that activation of OX40 signal by anti-OX40 agonistic antibody increased CD4-positive memory T cell induction, and anti-OX40 agonistic antibody also enhanced anti-tumor immune response [15,16].

As for its ligand, OX40 ligand (OX40L) was first identified as the protein gp34 expressed on T cells infected with human T-cell leukemia virus type 1 (HTLV-1) [17]. OX40L was later revealed to bind to OX40. OX40L is expressed on antigen-presenting cells such as activated B cells, dendritic cells, and Langerhans cells [18,19,20]. It has been reported that OX40 and OX40L are co-expressed in T cells infected with HTLV-1 [21]. With regard to MF/SS, OX40 was expressed by benign T cells and OX40L was expressed by c-Kit+ dendritic cells in MF [22]. However, the role of OX40 and OX40L in MF/SS has not been fully elucidated. The purpose of this study was to investigate the roles of OX40 and OX40L in MF/SS and to evaluate the possibility of these molecules as new therapeutic targets for MF/SS.

## 2. Results

### 2.1. OX40 and OX40L Expression in MF/SS Patients

We first examined mRNA expression levels of OX40 and OX40L in MF/SS lesional skin. OX40 and OX40L mRNA levels in lesional skin were significantly higher than those in normal skin (Figure 1A). In addition, patients with advanced stage MF and SS had higher OX40 and OX40L mRNA levels compared to healthy subjects (Figure 1B). Additionally, OX40 and OX40L mRNA expression levels positively correlated with serum levels of soluble IL-2 receptor in MF/SS patients (Figure 1C). Furthermore, OX40 mRNA expression positively correlated with OX40L mRNA expression in MF/SS lesional skin (Figure 1D). To evaluate the involvement of soluble forms of OX40L and OX40 in MF/SS, we measured serum OX40L and OX40 levels. Serum OX40L levels in MF/SS patients were below detection levels. On the other hand, the serum OX40 levels of MF/SS patients were detected, but they were equivalent to that of healthy controls (Figure 1E).

We next examined whether OX40 and OX40L mRNA levels correlated with patient survival using Kaplan–Meier analysis and log-rank test. A strong correlation was found between high OX40 mRNA levels and increased disease-related deaths (*p* = 0.04; log-rank test hazard ratio, 3.188) (Figure 1F). Similarly, MF/SS patients with high OX40L mRNA expression levels tended to exhibit poor prognosis compared to the low group, which did not reach statistical significance (*p* = 0.178; log-rank test hazard ratio, 2.411) (Figure 1F).

### 2.2. Immunohistochemical Staining of OX40 and OX40L in the Skin from MF/SS Patients

As OX40 and OX40L mRNA expression was increased in the MF/SS lesional skin, we next performed immunohistochemical staining of OX40L and OX40 using skin samples collected from patients with MF/SS. It was confirmed that OX40 and OX40L were expressed on tumor cells, including infiltrating lymphocytes in the epidermis, so-called Pautrier’s microabscess (Figure 2A). The number of OX40 or OX40L-positive cells in all stages of MF/SS skin was significantly higher than that of normal skin (Figure 2B).

### 2.3. Expression of OX40 and OX40L in Sézary Cells

To further confirm the increased expression of OX40 and OX40L in MF/SS, we performed flow cytometry using circulating tumor cells. In the peripheral blood of SS patients, CD4^+^CD7^−^ T cells, which were considered to be tumor cells, expressed both OX40 and OX40L, while CD4^+^ T cells in the peripheral blood of healthy subjects expressed OX40, but OX40L was weakly expressed (Figure 3). From these findings, MF/SS tumor cells co-expressed OX40 and OX40L, suggesting that overexpression of OX40L in tumor cells might be involved in the development of MF/SS.

### 2.4. Effects of OX40 and OX40L on CTCL Cell Lines

To further evaluate the effects of OX40 and OX40L on MF/SS, we used human CTCL cell lines. First, we analyzed the expression of both molecules on the cell surface by flow cytometry using three types of cell lines: HUT78 cell line (an SS cell line), MyLa cell line (an MF cell line), and HH cell line (an aggressive CTCL cell line). All three of these cell lines expressed both OX40 and OX40L on their cell surface (Figure 4A). It was confirmed that CTCL cells aberrantly expressed OX40L. Next, to assess the function of OX40–OX40L interactions in CTCL, we blocked OX40 and OX40L using neutralizing antibodies. Either anti-OX40 or anti-OX40L antibody significantly decreased the proliferation of CTCL cell lines, and the combination of both antibodies had a similar effect to single administration of either antibody (Figure 4B). In addition, anti-OX40 antibody induced apoptosis in CTCL cell lines (Figure 4C).

### 2.5. Examination of Intracellular Signal Changes Due to Inhibition of OX40–OX40L Interaction

OX40–OX40L signaling is known to induce phosphorylation of protein kinase B (AKT), extracellular signal-regulated kinase (ERK) 1/2, p38 mitogen-activated protein kinase (MAPK), and c-Jun N-terminal kinase (JNK) in various types of cells [23,24,25]. Consistent with previous reports, Western blot analysis showed that anti-OX40L antibody decreased phosphorylation of ERK1/2, p38 MAPK, and JNK in Hut78 cells (Figure 5A). In addition, longer the stimulation time tended to result in lower expression (Figure 5B). Based on these results, the anti-OX40L neutralizing antibody suppressed the phosphorylation of various intracellular signals.

### 2.6. Examination in a Tumor Growth Model Using Mice

Finally, to assess the in vivo effects of anti-OX40 and anti-OX40L antibodies, we used a xenograft model [26]. Treatment with the anti-OX40 or anti-OX40L antibody significantly decreased tumor formation by Hut78 cells in NSG mice in vivo (Figure 6A,B). There was no significant difference between the anti-OX40 group and the anti-OX40L group. In vivo experiments confirmed that tumorigenesis of MF/SS cells was suppressed by inhibiting the OX40–OX40L interactions.

## 3. Discussion

In this study, we revealed that MF/SS tumor cells co-expressed OX40 and OX40L. It has been reported that only OX40 is expressed on activated T cells and that OX40L is not or only slightly expressed on normal T cells [11,20]. Regarding OX40–OX40L interactions in hematological malignancies, it is known that T cells infected with HTLV-1 express both OX40 and OX40L [21]. It has also been reported that their expression is induced by a viral protein called Tax [21]. In adult T-cell leukemia/lymphoma, OX40 and OX40L stimulated adhesion of tumor cells to the vascular endothelium [27]. These previous reports suggested that co-expression of OX40 and OX40L in tumor cells would play an important role in tumor progression.

First, we showed that MF/SS tumor cells co-expressed OX40 and OX40L, which were involved in tumor growth and survival via various intracellular signals. OX40 and OX40L mRNA expression in the lesional skin was increased as the stages progressed (Figure 1B). Consistently, expression of OX40 and OX40L was positively correlated with serum sIL-2R levels, a representative MF/SS disease marker (Figure 1C). Furthermore, survival analysis showed that patients with elevated OX40 mRNA expression had an increased risk of disease-related mortality (Figure 1F). Similarly, peripheral blood mononuclear cells from patients with SS and healthy individuals were collected, and flow cytometry was used to analyze the expression of OX40 and OX40L on the cell surfaces. Circulating Sézary cells aberrantly expressed OX40L, while normal T cells did not express OX40L (Figure 3). In addition, in vitro experiments confirmed the expression of OX40 and OX40L in human CTCL cell lines.

Next, to evaluate in detail the function of OX40–OX40L interactions in CTCL cell lines, we used a neutralizing antibody to inhibit OX40 and OX40L. Inhibition of OX40 and OX40L suppressed tumor cell proliferation, and in terms of viability, the OX40 neutralizing antibody induced apoptosis of CTCL cells (Figure 4C). Regarding OX40–OX40L signaling, it has been reported that the downstream of OX40–OX40L is PI3-kinase/AKT signaling pathway, FAS signaling pathway, and NF-κB and Bcl-2/Bcl-xl activation [17,18,19]. When we analyzed the changes in phosphorylation of signal transduction substances by the Western blotting method, the expression of phosphorylated ERK1/2, phosphorylated p38 MAPK, and phosphorylated JNK were reduced by the anti-OX40L neutralizing antibody (Figure 5B). From the above results, it was considered that OX40 and OX40L were co-expressed in MF/SS tumor cells and that the tumor cells proliferated via phosphorylation signals due to the interaction between OX40 and OX40L on tumor cells. Finally, we evaluated the in vivo effects of anti-OX40 and anti-OX40L neutralizing antibodies in a tumor dissemination model using immunodeficient mice. Tumor formation of Hut78 cells in immunodeficient mice was significantly suppressed in the anti-OX40 and anti-OX40L neutralizing antibodies-administered groups as compared with the control group (Figure 6). It is already known that OX40 and OX40L are co-expressed in HTLV-1-infected T cells. Similarly, by acquiring OX40L expression in the process of tumorigenesis, MF/SS tumor cells promote the growth of the tumor itself.

Recently, it was revealed that the interaction between CD137, which belongs to the same TNF receptor superfamily as OX40, and its ligand, CD137L, favored tumor progression in MF/SS [28]. CD137, normally expressed on the surface of activated T cells and NK cells, was overexpressed on tumor cells in MF/SS [19]. CD137L was also expressed on tumor cells and suggested that the interaction of CD137 and CD137L promoted tumor cell proliferation, survival, migration, and tumorigenesis and might be a promising therapeutic target for MF/SS [28,29,30,31]. Similarly, in this study, it was clarified that the interaction between OX40 and OX40L is greatly involved in the pathophysiology of MF/SS, which led us to expect that anti-OX40 and anti-OX40L antibodies could be promising therapeutics for MF/SS.

Based on the fact that the OX40 signal enhances anti-tumor effects on T cells and NK cells, the OX40 agonist antibody has been recently used as an anti-tumor drug to various malignancies. Concretely, clinical trials of OX40 agonist antibody drugs, such as MOXR09016, dacetuzumab, CP-870, CP-893, and Chi Lob 7/4, have been planned and started for non-small cell lung cancer, squamous cell carcinoma of the head and neck, malignant melanoma, triple-negative breast cancer, and colorectal cancer [32,33]. However, the effects of such immunotherapy varied greatly from patient to patient, and the actual therapeutic effects were only seen in some patients. One of the causes is that the effect of immunotherapy depends on the state of tumor immunity of each patient. Therefore, at this point, combination therapy using several immunotherapies and standard treatments is recommended. Evaluating the state of tumor immunity to identify the group of patients who are expected to respond to immunotherapy would be necessary. To summarize, OX40 agonists have been used as anti-neoplastic drugs in the above malignancies. On the other hand, in MF/SS, unlike other malignancies, OX40 and OX40L were co-expressed in MF/SS tumor cells (Figure 7). OX40-OX40L interactions are thought to be associated with tumorigenesis by a mechanism different from the tumor immune response in MF/SS. Accordingly, disruption of OX40-OX40L interactions would be beneficial in the treatment of MF/SS. A treatment strategy which enhances anti-tumor effects by the OX40 agonist may involve risks of activating tumor cell growth in MF/SS whereas blocking OX40-OX40L would inhibit tumor growth. In fact, our in vivo experiment showed that blocking OX40 or OX40L significantly suppressed tumor growth. Here, our in vivo experiment had some limitations. NSG mice are immunodeficient, and the treatment by blocking OX40 or OX40L involved activation of the immune system. Further experiments would be needed to confirm the therapeutical potential by disruption of OX40–OX40L in MF/SS.

Taken together, we showed that MF/SS tumor cells co-expressed OX40 and OX40L. As OX40–OX40L interactions contributed to the proliferation of MF/SS tumor cells, disruption of these interactions could be a new therapeutic strategy for the treatment of MF/SS.

## 4. Materials and Methods

### 4.1. Patients and Tissue Samples

Eighty-five patients with MF/SS (56 male and 29 female patients; mean ± SD age: 59.1 ± 14.3 years; 21 cases of patch MF, 20 cases of plaque MF, 30 cases of tumor MF, and 14 cases of SS) and twenty healthy controls (45.8 ± 19.0 years) were enrolled in this study. Skin samples were collected from 65 patients with MF/SS (32 cases of patch and plaque MF, 25 cases of tumor MF, and 8 cases of SS; 43 males and 22 females; mean ± SD age: 59.6 ± 14.6 years), and 13 healthy controls (53.6 ± 17.8 years). Blood samples were collected from 72 patients with MF/SS (53 male and 19 female patients; 59.9 ± 15.1 years; 16 cases of patch MF, 15 cases of plaque MF, 28 cases of tumor MF, 3 cases of erythrodermic MF, and 10 cases of SS) and 16 healthy controls (53.3 ± 14.1 years). As previously described [14], peripheral blood mononuclear cells (PBMCs) were obtained from 6 SS patients and 6 healthy controls. All patients with MF and SS were diagnosed according to ISCL/EORTC criteria [1,15]. All samples were collected after informed consent during daily clinical practice. The medical ethical committee of the University of Tokyo approved all described studies (0695-17), and the study was conducted according to the Declaration of Helsinki Principles.

### 4.2. Cell Lines

HH cells (an aggressive CTCL cell line), Hut78 cells (an SS cell line), and MyLa cells (an MF cell line) were kind gifts from Dr. Kazuyasu Fujii (Department of Dermatology, Kagoshima University, Kagoshima, Japan). Hut78, HH, and MyLa cells were cultured in RPMI 1640 with 10% FBS and supplements (penicillin G sodium, streptomycin sulphate, and amphotericin B).

### 4.3. Quantitative RT-PCR

RNA was extracted from human skin samples with RNeasy Fibrous Tissue Mini Kit (Qiagen, Valencia, CA, USA). Total RNA was extracted from CTCL cell lines with TRIzol Reagent (Invitrogen, Carlsbad, CA, USA). Quantitative RT-PCR was performed based on the SYBR Green assay. The mRNA levels were normalized to those of the GAPDH gene. The relative change in the levels of genes of interest was determined by the 2^−ΔΔCT^ method. Primers are OX40 forward, 5′-TCA GAA GTG GGA GTG AGC GGA AG-3′, reverse, 5′-GCA GAG AGC CGG AGG CAG CCA TCG GC-3′; OX40L forward, 5′-TGC TTC ACC TAC ATC TGC CTG CA-3′, reverse, 5′-CTA GTA GGC TCA AGG CAA TCT TG-3′.

### 4.4. Survival Analysis

We divided CTCL patients into two groups by setting cutoff values of OX40 and OX40L mRNA in the lesional skin and compared overall survival between the two groups. The best discriminating cutoff value of each factor was established using receiver operating characteristic analysis. The cutoff values of OX40 and OX40L mRNA expression levels were defined as 0.007 and 0.002, respectively.

### 4.5. Immunohistochemistry

We performed immunohistochemical staining for OX40L and OX40 using skin of MF and SS (*n* = 20; 13 male and 7 female patients; mean ± SD age: 56.9 ± 13.2 years; 5 cases of patch MF, 4 cases of plaque MF, 5 cases of tumor MF, and 6 cases of SS) and normal skin (*n* = 7; mean ± SD age: 28.8 ± 6.4 years). These sections were stained with either sheep anti-human OX40L polyclonal antibody (LS-B14343; LSBio, Seattle, WA, USA), rabbit anti-human OX40 polyclonal antibody (R&D Systems, Minneapolis, MN, USA) or isotype-matched control antibodies followed by ABC staining (Vector Lab, Burlingame, CA, USA). Each section was examined independently by three investigators in a blinded manner.

### 4.6. Enzyme-Linked Immunosorbent Assay

Serum levels of soluble OX40L and soluble OX40 were quantified using the human soluble OX40L ELISA Kit (Abcam, Cambridge, MA, USA) and human soluble OX40 ELISA Kit (Thermo Fisher Scientific, Waltham, MA, USA), according to the manufacturer’s instructions. These assays were based on the quantitative sandwich enzyme immunoassay technique.

### 4.7. Flow Cytometric Analyses

We performed flow cytometric analyses using CTCL cell lines and PBMCs from SS patients as well as healthy controls. For isolation of CD4^+^ cells, PBMCs were stained with anti-CD4-PE antibody (RPA-T4; Biolegend) followed by secondary antibody BD IMag anti-PE magnetic particles-DM (BD Biosciences, San Diego, CA, USA), and targeted cells were isolated using a BD IMagnet (BD Biosciences). Antibodies used are listed anti-human OX40-PE-Cy7 (clone ACT35; Biolegend, San Diego, CA, USA), anti-human OX40L-PE (clone 11C3.1; Biolegend), anti-human CD4-APC (clone 13B8.2; Beckman Coulter, Miami, FL, USA), anti-human CD7-FITC (clone 8H8.1; Beckman Coulter). Negative staining was defined by replacing the antibody with its isotype-matched control. FACScan flow cytometer and Cell-Quest software (BD Biosciences) were used.

### 4.8. Western Blotting

Protein extract preparation and Western blot analysis were performed as previously described using antibodies. Primary antibodies included AKT, phosphorylated AKT, ERK 1/2, phosphorylated ERK 1/2, p38 MAPK, phosphorylated p38 MAPK, JNK, phosphorylated JNK (Cell Signaling Technology, Beverly, MA, USA), and β-actin (Santa Cruz Biotechnology, Santa Cruz, CA, USA). The density of each band was quantified with ImageJ software (National Institutes of Health, Bethesda, MD, USA).

### 4.9. Proliferation Assays by Cell Count

Hut78 cells, HH cells, and MyLa cells were co-cultured with anti-OX40 or anti-OX40L neutralizing antibody and then stained with trypan blue to count the number of surviving cells. Anti-human OX40 (2 μg/mL; AF3388; R&D Systems) and anti-human OX40L (2 μg/mL; clone MM0505-8S23; Novus Biologicals, Centennial, CO, USA) were used as neutralizing antibodies.

### 4.10. Apoptosis Assays

Apoptosis was analyzed using Annexin V (conjugated FITC or PE) and 7-amino-actinomycin D (7-AAD) provided by Biolegend, according to the manufacturer’s instructions.

### 4.11. In Vivo Animal Experiments

Hut78 cells (5.0 × 10^6^ cells) in 100 µL of PBS were injected subcutaneously into the shaved left abdomen of NOD/SCID interleukin-2 receptor γ-chain-deficient (NSG) mice obtained from Charles River Laboratories. On days 0, 4, 7, and 11, we intraperitoneally injected the human OX40L neutralizing antibody (40 µg/mL; clone MM0505-8S23; Novus Biologicals) in 100 µL of PBS in the anti-OX40L antibody-treated group. Similarly, human OX40 neutralizing antibody (40 µg/mL; AF3388; R&D Systems) in 100 µL of PBS was used in the anti-OX40 antibody-treated group, whereas the control group was injected with isotype IgG (R&D systems). The tumor volume was calculated using the equation: *V* = π (L1 × L2^2^)/6, where *V* = volume (mm^3^), L1 = longest diameter (mm), and L2 = shortest diameter (mm).

### 4.12. Statistical Analysis

GraphPad Prism 7.01 software program was used for statistical analyses. All in vitro experiments were repeated at least 3 times and mean ± SD was determined. Statistical analysis between 2 groups was performed using the Welch’s *t* test. For comparing two group values that did not follow Gaussian distribution, a two-tailed Mann–Whitney *U* test was used. A paired *t* test was used to determine significant differences between anti-OX40L neutralizing antibody-treated and control groups when using patient tumor cell samples. Correlation coefficients were determined by using the Spearman’s rank correlation test. Disease-specific survival was estimated by the Kaplan–Meier method, and the differences in survival between the two groups were assessed by the log-rank test. *p* values of <0.05 were considered statistically significant.

## Figures and Tables

**Figure 1 ijms-22-12576-f001:**
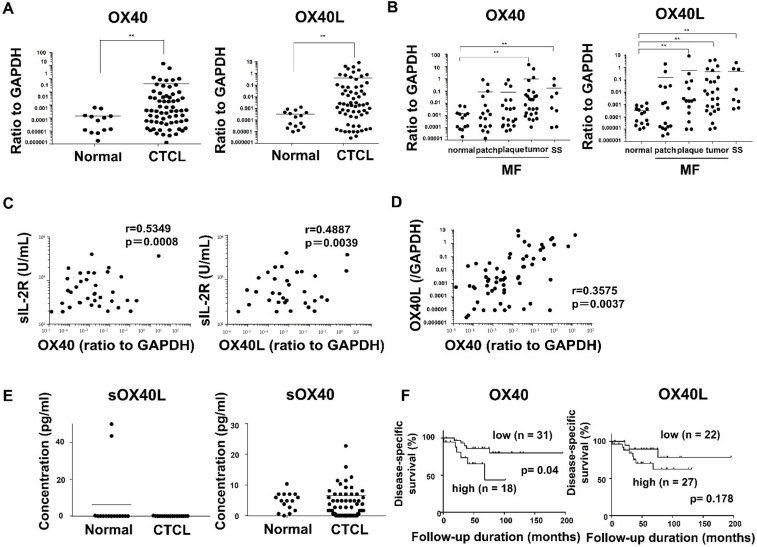
OX40 and OX40 ligand (OX40L) expression in patients with cutaneous T-cell lymphoma (CTCL). (**A**,**B**) Quantitative RT-PCR was performed to measure expression levels of OX40 and OX40L using mRNA extracted from MF/SS lesional skin (*n* = 57; 49 MF cases and 8 SS cases) and normal skin (*n* = 13). (**C**) Correlations between soluble IL-2 receptor (sIL-2R) and OX40 or OX40L mRNA expression in MF/SS patients. (**D**) Positive correlations between OX40 and OX40L mRNA expression in MF/SS lesional skin. (**E**) Serum soluble OX40 and soluble OX40L levels in CTCL patients. (**F**) MF/SS patients with high OX40 mRNA expression exhibited poor prognosis. Means are presented as bars. ** *p* < 0.01.

**Figure 2 ijms-22-12576-f002:**
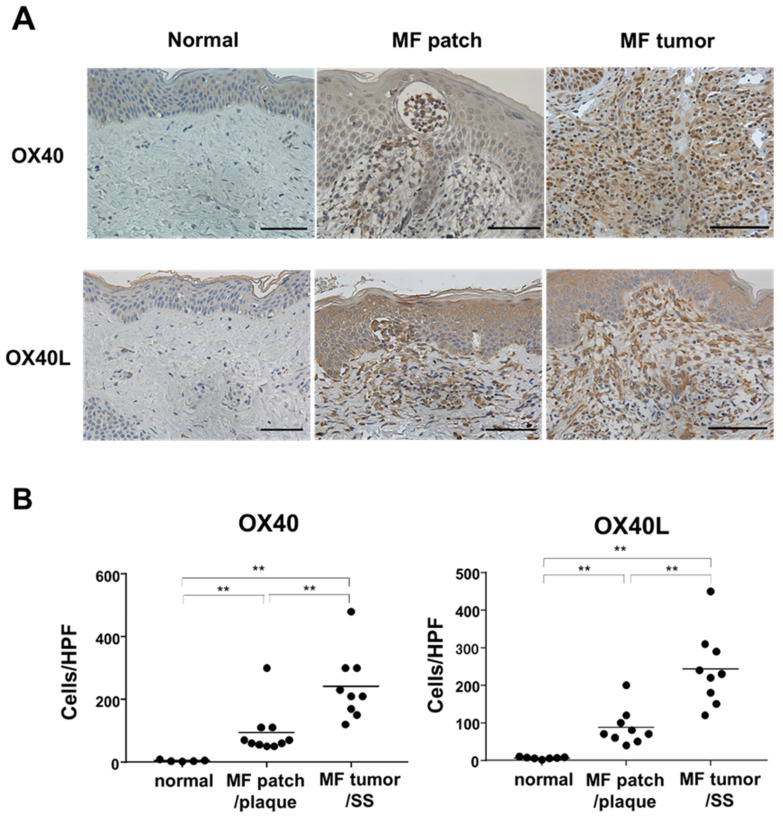
OX40 and OX40L expression in lesional skin of MF/SS was determined by immunohistochemistry (original magnification ×400, scale bar = 100 µm). (**A**) Representative results are shown. (**B**) Counts of OX40 and OX40L-positive cells. Ten cases of patch/plaque MF, nine cases of tumor MF/SS, and seven cases of healthy subjects. ** *p* < 0.01.

**Figure 3 ijms-22-12576-f003:**
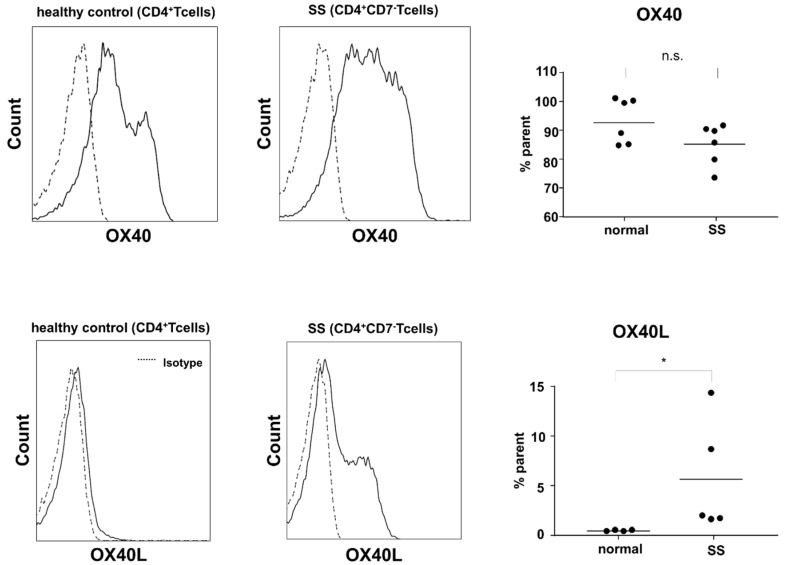
OX40 and OX40L expression was analyzed by flow cytometry in CD4^+^CD7^−^ T cells from 6 Sézary syndrome (SS) patients and CD4^+^ T cells from 6 healthy controls. Comparison of OX40 and OX40L-positive cell ratios in peripheral blood CD4^+^CD7^−^ T cells in patients with SS or in healthy controls peripheral blood CD4^+^ T cells. n.s.: not significant. * *p* < 0.05.

**Figure 4 ijms-22-12576-f004:**
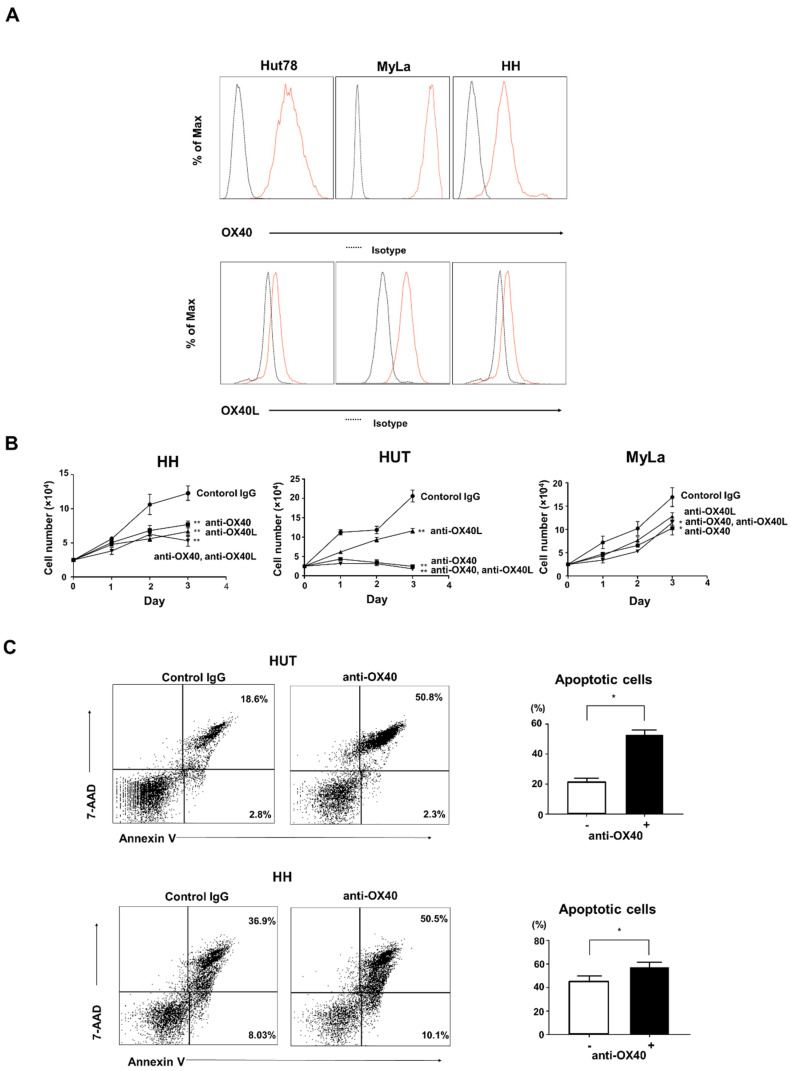
Involvement of OX40 and OX40L in CTCL cell lines. (**A**) OX40 and OX40L expression was analyzed by flow cytometry in human CTCL cell lines (Hut78, MyLa, and HH cells). (**B**) HH, Hut, and MyLa cells (1.0 × 10^5^/well) were cultured with anti-OX40 (5 μg/mL) and/or anti-OX40L (5 μg/mL) neutralizing antibodies for 48 hours. (**C**) Hut and HH cells (1.0 × 10^5^/well) were cultured with anti-OX40 neutralizing antibody (5 μg/mL) for 24 hours. Apoptosis (Annexin V^+^ and 7-AAD^−^) was evaluated with Annexin V and 7-AAD staining for flow cytometric analysis. * *p* < 0.05, ** *p* < 0.01.

**Figure 5 ijms-22-12576-f005:**
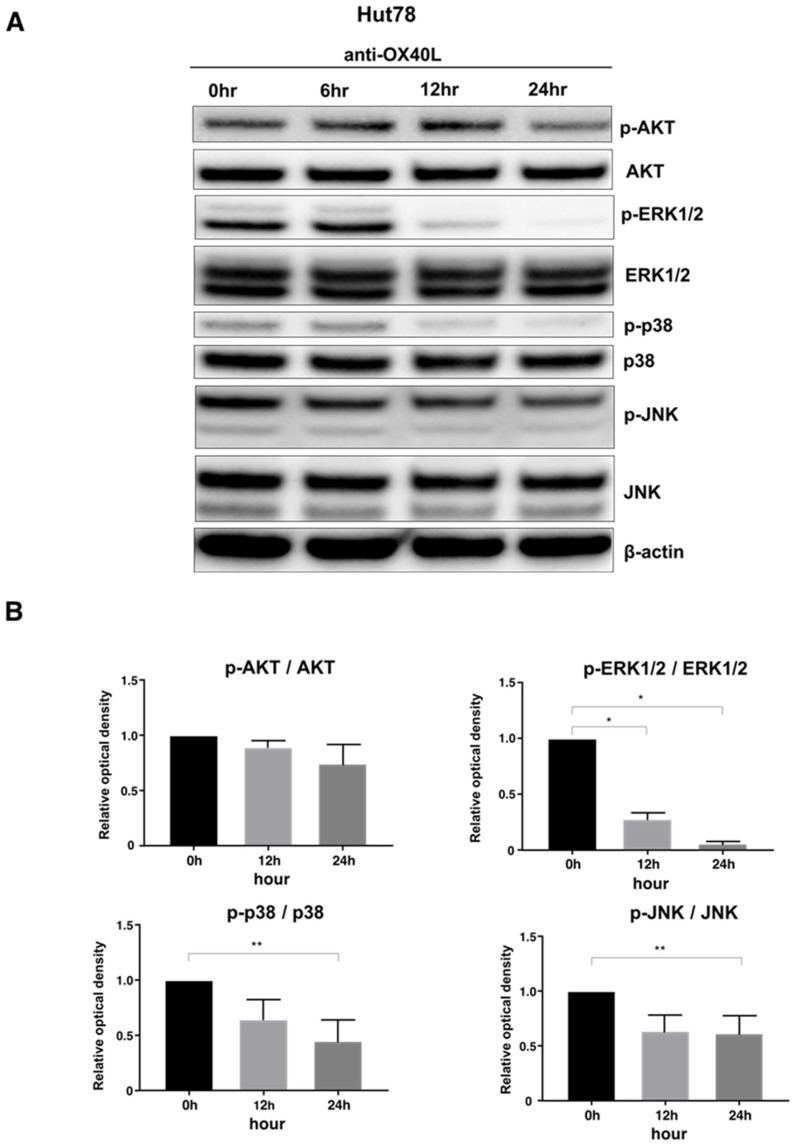
Western blotting analysis was conducted on the lysates of Hut78 cells. (**A**) Hut78 cells were treated with anti-OX40L neutralizing antibody (5 μg/mL) or isotype control for 0, 6, 12, or 24 h. Phosphorylation of AKT, ERK1/2, p38 MAPK, and JNK were measured. (**B**) Changes in phosphorylation of various signal transduction substances. * *p* < 0.05, ** *p* < 0.01.

**Figure 6 ijms-22-12576-f006:**
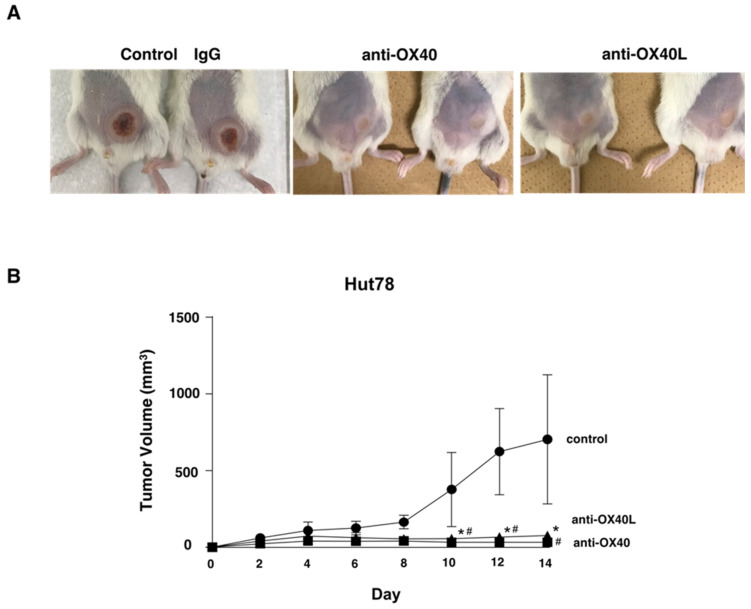
Anti-OX40 and anti-OX40L antibodies suppressed tumor growth. Hut78 cells (5.0 × 10^6^ cells) in 100 µL of PBS were injected subcutaneously into the shaved left abdomen of NSG mice. Anti-OX40 or anti-OX40L neutralizing antibodies were injected on days 0, 4, 7, and 11 intraperitoneally. (**A**) Representative images are shown. (**B**) The tumor volume was calculated using the equation: *V* = π (L1 × L2^2^)/6, where *V* = volume (mm^3^), L1 = longest diameter (mm), and L2 = shortest diameter (mm). Values are means ± SEM. *,# *p* < 0.05 by Mann–Whitney *U* test compared with control group.

**Figure 7 ijms-22-12576-f007:**
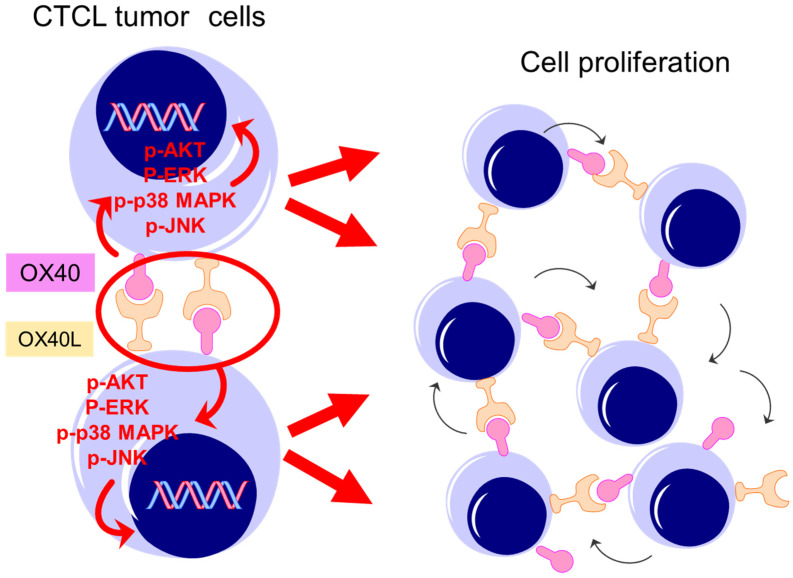
Schematic model of the main findings of this study.

## Data Availability

The data presented in this study are available on request from the corresponding author. The data are not publicly available due to privacy.

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
