# Peer review of "Roles of OX40 and OX40 Ligand in Mycosis Fungoides and Sézary Syndrome"

_ijms, 2021, doi:10.3390/ijms222212576_

Round 1
Reviewer 1 Report
They studied OX40 and OX40L expression and function using clinical samples of MF and SS and CTCL cell lines. They found that OX40 and OX40L were expressed on tumor cells of MF and SS. Anti-OX40 antibody and/or anti-OX40L antibody suppressed the proliferation of CTCL cell lines both in vitro and in vivo.
This is an interesting study.
There are a few concerns:
- The description of Figure 1D is confusing – the text says that it is the serum OX40 and OX40L, but the legend says skin lesion. If it is serum level, it does not seem to make sense in the context of Figure 1E (left panel).
- Figure 1F: what is the source of the mRNA? Tumor cells isolated from the skin lesion? or the whole skin lesion without tumor isolation? Please describe this in the text.
- Description of Figure 3: there is no significant difference between normal vs. ss with regard to OX40, so the statement in the text is not accurate “…suggesting that overexpression of OX40L and OX40 in tumor cells might be involved in the development of CTCL” – OX40 is expressed but not overexpressed.
- Figure 6 legend: “Hut78- anti-OX40L or Hut78- anti-OX40 cells” – what does this mean? Does it mean that the Hut78 cells were pre-mixed with anti-OX40 or anti-OX40L antibodies then subcu grafted?
- Methods:
Section 4.11. In vivo animal experiments
“In other experiments, NSG mice were injected subcutaneously into the shaved left abdomen with transduced Hut78 cells..” - what are “transduced Hut78 cells”? Transduced by what? Is the data shown in Figure 6 the transduced Hut78 or Hut78?

Reviewer 2 Report
The manuscript “Roles of OX40 and OX40 ligand in cutaneous T-cell lymphoma” by Kawana et al is an interesting and well-presented piece of work. Authors postulate that OX40 and OX40L are abnormally expressed in cutaneous T cell lymphoma (CTCL) and that it may be related to the disease progression. In my opinion, the manuscript is clear and well-organized. My main criticism would concern the way some passages are written (see below). However, as far as the data presentation is concerned, I think the quality of their work is satisfactory.
Minor comments
Theoretical part. I think I would not mind reading a bit more about the physiological functions of OX40 and OX40L. Meanwhile, even the synonym (CD134) is lacking. Also, there are indeed very few publications dealing with OX40(L) and CTCL, which means it is even more important to carefully report the current state of art. Perhaps it would be a good idea to cite the work by Vieyra-Garcia et al., 2019 (PMID: 30626755).
Discussing the findings in the context of the functioning of the immune system is also necessary. The shortcomings of the animal model used by the Authors should be addressed. NOD/SCID mice are immunodeficient, and the treatment proposed by the Authors involves activation of the immune system. In most works cited by the Authors, OX40 agonists have the anti-neoplastic effect, while the Authors would see a therapeutical potential rather in inactivation OX40 and disruption of 0X40/OX40L interaction. I think that requires a more detailed commentary.
Page 2. “It is already known that cell to cell interactions between tumor cells by autocrine or paracrine contribute to survival and growth in CTCL”. Only tumor cells or perhaps interactions between tumor cells and/or tumor cells and tumor microenvironment? Also “paracrine” and “autocrine” are adjectives, so something is missing. Paracrine signaling?
Figure 1. Please check the figure legend and explanation in the text. Fig 1C is (according to the Results, 2.1) presenting “serum OX40 and OX40L levels”, however, the X axis is labeled “ratio to GAPDH”. Also, according to Fig. 1E serum levels of OX40L are barely detectable, which seems to be contradictory to findings in Fig. 1C.
Fig 1D, according to the body text, shows serum levels of OX40 and OX40L, but according to the figure legend it is actually expression in lesional skin. I suggest the Authors go once again carefully through this section and verify which data refers to what.
Immunohistochemical staining… (chapter 2.2). How many patient samples were used? Are these the same patients as in Fig. 1 (chapter 2.1)?
Figure 4. Cell viability data are not shown for MyLa. Was this analysis performed? Was it omitted, because the results were not significant?
Chapter 2.7 Results for OX40L are described. How about OX40?
Figure 6 legend. “Hut78- anti-OX40L or Hut78- anti-OX40 cells were injected (…)” Hut78- anti-OX40 cells?
The way the data is presented in figure panels. I personally find it very confusing that sometimes OX40 comes first and sometimes OX40L. I suggest that the Authors keep one of them (preferably OX40) consistently as the first one.
Statistics
Choice of SD versus SEM. “All in vitro experiments were repeated at least 3 times and mean ± SD was determined” but then, in the description of fig. 6, regarding the animal study: “Values are means ± SEM”. Why is that? Also, I think giving SEM is more correct here, as we try to estimate the real mean, and the spread (error) gets smaller as we collect more data. Standard deviation (SD), in the meantime, makes perfect sense in case of a spread in a known population (such as the age of subjects used in the study).
Language
The manuscript is mostly written in a clear way, but there are some passages not very fortunately formulated, containing unnecessary repetitions etc. Perhaps a check by an English native speaker would be beneficial.
Specifically:
“OX40 was first reported as a surface protein of activated CD4- positive T cells in 1987 [11], which was also reported to be expressed on T cells (…)”. Which…?
“ (…) anti-OX40 agonistic antibody blocked inhibitory function of Tregs and main[1]tained the proliferation and function of effector T cells using peripheral blood of healthy subjects in vitro level. “ Has it been demonstrated in vitro using peripheral blood? This way it actually sounds as if OX40 agonistic antibody performed its function using peripheral blood of healthy subjects and it is absolutely not clear what the “in vitro levels” refer to.
“It has been suggested that Tregs are targets of anti-OX40 antibody as well as effector T cells.” (..) Tregs, as well as effector T cells, are targets…?
“…and we expected that anti-OX40 and anti-OX40L antibodies could be promising therapeutic targets for CTCL.” I think a target (as correctly mentioned just above) can be an interaction (here: interaction of CD137-CD137L) or a structure. Antibodies might be therapeutics, they may be used in the treatment of…
“Evaluating the state of tumor immunity to identify the group of patients who are expected to be effective in immunotherapy would be necessary” – patients to be effective?
Chapter 2.5: “OX40L and OX40 expression was analyzed by flow cytometry in CD4+CD7- T cells from 6 Sézary syndrome (SS) patients, CD4+ T cells from 6 healthy controls. Comparison of OX40L and OX40-positive cell ratios in peripheral blood CD4+CD7- T cells in patients with SS or in healthy controls peripheral blood CD4+ T cells.” – it is unclear what this fragment refers to.
“(…) we injected human OX40L neutralizing antibody (40 µg/mL; clone MM0505-8S23; Novus Biologicals) or hu[1]man OX40 neutralizing antibody (40 µg/mL; AF3388; R&D Systems) in 100 µL of PBS to anti-OX40L, anti-OX40 antibody group-mice intraperitoneally.” I think I understand what the Authors mean, but the syntax of this sentence is slightly off.
Also, 2.1, 2.2 and 2.3 basically all start with “expression of OX40 and OX40L was evaluated”. The opening sentence should preferably vary a bit and give information about what the following section is supposed to describe (for example it may refer to a different level i.e. mRNA versus protein, or different stage of the disease, or at least “further corroborates”).
Reviewer 3 Report
- There are several distinct types of cutaneous T-cell lymphoma (CTCL) other than mycosis fungoides (MF) and Sezary syndrome (SS). Since the subjects of this study are MF and SS, the article title and manuscript text should be revised accordingly. CTCL is not equivalent to MF/SS.
- Page 2, Line 16. .. a mouse model “shown” … The “shown’ should be revised to “showed”.
- Page 2. The last paragraph on OX40 ligand should be revised. Many of the sentences are too long and unclear. Please revise them to more simple sentences to make it clear.
- In Figure 3, the data on OX40 should be on top of that for OX40L, a consistency in the order of appearance as in other Figures and text, similarly in Panel 4A.
- Page 4. Last 3rd “.. slightly expressed”. …” It might be better to use “dimly’ or “weakly” to replace “slightly”.
- Page 5. Last 3rd “… CTCL cells ectopically expressed …” It might be better to use “aberrantly” to replace “ectopically”. Similarly, in the first line of Discussion and other parts.
- Page 10. 3rd paragraph about CD137 and CD137L. The authors wrote “In this study, it was clarified that the interaction ….” This sentence has to be rephrased for clarification. Did the authors imply that there is an interaction between OX40-OX40L and CD137-CD137L?
- Page 11. The 4th A period is needed after CTCL.
